# Probabilistic Semantic Mapping for Autonomous Driving in Urban Environments

**DOI:** 10.3390/s23146504

**Published:** 2023-07-18

**Authors:** Hengyuan Zhang, Shashank Venkatramani, David Paz, Qinru Li, Hao Xiang, Henrik I. Christensen

**Affiliations:** Autonomous Vehicle Laboratory, Contextual Robotics Institute, University of California San Diego, La Jolla, CA 92093, USA; svenkatramani@ucsd.edu (S.V.); dpazruiz@ucsd.edu (D.P.); q4li@ucsd.edu (Q.L.); haxiang@g.ucla.edu (H.X.); hichristensen@ucsd.edu (H.I.C.)

**Keywords:** autonomous vehicles, semantic mapping, semantic segmentation, fusion

## Abstract

Statistical learning techniques and increased computational power have facilitated the development of self-driving car technology. However, a limiting factor has been the high expense of scaling and maintaining high-definition (HD) maps. These maps are a crucial backbone for many approaches to self-driving technology. In response to this challenge, we present an approach that fuses pre-built point cloud map data with images to automatically and accurately identify static landmarks such as roads, sidewalks, and crosswalks. Our pipeline utilizes semantic segmentation of 2D images, associates semantic labels with points in point cloud maps to pinpoint locations in the physical world, and employs a confusion matrix formulation to generate a probabilistic bird’s-eye view semantic map from semantic point clouds. The approach has been tested in an urban area with different segmentation networks to generate a semantic map with road features. The resulting map provides a rich context of the environment that is valuable for downstream tasks such as trajectory generation and intent prediction. Moreover, it has the potential to be extended to the automatic generation of HD maps for semantic features. The entire software pipeline is implemented in the robot operating system (ROS), a widely used robotics framework, and made available.

## 1. Introduction

Many approaches to design of autonomous vehicles rely on high-definition (HD) maps to model the static parts of the environment. These maps provide crucial information such as centimeter-level definitions of road networks, traffic signs, crosswalks, traffic lights, and speed limits. Due to the dynamic nature of the real world, these maps can quickly become outdated, especially during road network changes or construction. Manually annotating HD maps is a laborious and time-consuming process, and outdated maps can lead to unsafe scenarios when vehicles perform inadequate reference path tracking actions. Extracting semantics and attributes from data is one of the most challenging aspects of HD map generation [1]. Given this, a method that automates semantic extraction could significantly improve HD map generation, reduce labor costs, and enhance driving safety.

Generating centimeter-level semantic labels for a scene is a cumbersome task. Many efforts approach this problem from the perspective of scene understanding. Prior work has used conditional random fields (CRF) to assign semantic labels [2,3]. More recently, deep learning techniques have shown promising results in retrieving semantic information from images [4,5,6], point clouds [7], or both [8]. However, semantic scene understanding does not account for stitching together individual observations to generate a map representation.

Some researchers have also explored methods to create semantic maps of the environment, including [9,10,11]. However, these approaches either rely on aerial imagery/high-cost sensors to extract road information, which can limit the availability of data, or they do not explicitly map lane and crosswalk information, which are crucial for HD map generation.

Other work directly generates the lane-level HD map [7,12,13,14,15] or topology map [16]. These maps are in sparse vectorized representation that can be valuable for planning. However, these methods are limited by a small set of map elements. For this reason, the generated maps lack the rich context required for urban driving.

Our study addresses gaps in the automatic generation of dense probabilistic semantic maps in urban driving environments. To achieve this, we propose a semantic mapping pipeline that creates a bird’s-eye view (BEV) semantic map of the environment instead of a single-frame semantic understanding. The pipeline utilizes a confusion matrix to incorporate the uncertainty of the semantic segmentation network into mapping and fuses light detection and ranging (LiDAR) intensity to map lane marks accurately. We leverage dense point maps obtained from a 16-channel LiDAR to reduce the cost and increase data availability. Furthermore, our work builds on state-of-the-art semantic segmentation networks [6,17] that are trained exclusively on publicly available datasets [18], providing rich semantic labels, including roads, lane marks, crosswalks, and sidewalks. To evaluate the effectiveness of the proposed model, we compare it with ground-truth HD maps generated for our campus and use data from our autonomous vehicle. The results demonstrate that our model accurately identifies semantic features on the road and can effectively map them with a small error margin.

We augmented our initial work [19] by adding new semantic segmentation models and adding extensive analysis with modified precision and recall, which are more appropriate for evaluating mapping performance. We additionally open source the code for running our entire pipeline.

The paper is organized with an initial discussion of related work in Section 2. We present the overall methodology in Section 3 and the associated experiments in Section 4. Based on our results, there are a number of issues to consider regarding standard datasets, labeling, and evaluation, which are discussed in Section 5 before we summarize in Section 6.

## 2. Related Work

In this section, we will briefly summarize related work across the areas of segmentation (Section 2.1), mapping (Section 2.2), HD map generation (Section 2.3), and probabilistic maps (Section 2.4).

### 2.1. Semantic Segmentation

There has been significant progress in the field of semantic segmentation, which involves assigning semantic labels to each data point (e.g., pixel or voxel). Large-scale datasets like CityScapes [20], CamVid [21], and Mapillary [18] have accelerated this progress in the domain of road scenes. Semantic segmentation algorithms that provide pixel-level information can be particularly useful for building HD maps, which require fine-grained labeling for scene objects.

2D semantic segmentation approaches, such as those in [4,5,22], use encoder-decoder architectures to interpret global and local information in images. These models, when trained on the aforementioned large datasets, can effectively segment objects on the road. 3D semantic segmentation approaches have utilized convolutional neural networks (CNNs) to classify points in LiDAR point clouds after a transformation into range images, in [23,24,25]. These methods provide promising results but fail to distinguish objects with textural differences. Full 3D semantic segmentation using voxel-based approaches has also been proposed [26]; however, it requires 3D convolutions on dense raw point clouds (32 or 64 channel LiDARs), making real-time operation challenging. New transformer-based approaches have shown improvements in evaluation metrics [27], though they require higher computational capabilities for full self-attention. Recent work has also tried to directly generate semantic segmentation in BEV from a single image [28] or using the fusion of LiDAR and a camera [8].

### 2.2. Semantic Mapping

The term semantic mapping has taken various meanings in literature [29]. For our purposes, we have chosen to follow the definition provided in [30], which is a map that contains environmental attributes and occupancy metrics. For the task of autonomous vehicles, this encompasses features such as drivable areas and road features.

There are alternative methods that utilize CRF-based techniques to achieve high-density semantic mapping [3]. In this instance, an associative hierarchical CRF is utilized for semantic segmentation, while a pairwise CRF is used for mapping. The latter strategy ensures that the output remains smooth. Another approach, detailed in [31], involves using a stereo pair to estimate depth reliably. However, this particular method does not account for the explicit mapping of crosswalks and lanes, both of which are necessary for the creation of HD maps.

In a related study, Maturana et al. [9] combine semantic imagery captured by a camera with LiDAR point clouds. They rely on raw point clouds in real time from a 64-channel LiDAR, which provides more dense real-time information at a higher cost. Our approach, however, can build a map from a relatively cheaper 16-channel LiDAR. Moreover, their research concentrates on off-road environments, whereas our research focuses on urban driving scenarios. In such settings, certain traffic rule-related categories, like crosswalks and lane markings, require higher attention.

### 2.3. HD Map Generation

The generation of HD maps has been explored from various perspectives, including online and offline mapping. Zhou et al. [12] propose to construct lane-level HD maps for urban environments. They first use cameras and LiDARs for 3D semantic reconstruction and then use the OpenStreetMap (OSM) with a semantic particle filter to generate offline lane-level HD maps for the urban environment.

Online methods are gaining popularity. Homayounfar et al. [7] generate a lane-level map for the highway, which is facilitated by large-scale open datasets with HD map data such as nuScenes [32]; Argoverse 2 [33,34]; and OpenLane-V2 [35], a line of work focused on generating online HD map for urban environments. Li et al. [36] propose HDMapNet, which generates rasterized maps, while Liu et al. [13] propose VectorMapNet to generate vectorized representations directly. MapTR [14] and TopoNet [15] improve mapping performance by using permutation invariant representations and a topology-preserving loss, respectively. Can et al. [16] propose a loss that captures the accuracy in estimating topology. Additionally, HD maps can be built from aerial imagery [37], but the availability of data can present a limitation. These works focus on sparse lane-level representations with predefined map element types. In contrast, our generated dense maps can capture all of the semantic classes from the semantic segmentation network.

### 2.4. Probabilistic Map

Probabilistic mapping builds a map that maximizes the likelihood of the map under the data [38]. Thrun et al. [38] build a probabilistic map by modeling the occupancy probability with expectation maximization. Their work and many other works [39,40] address simultaneous localization and mapping (SLAM), while our work focuses only on the mapping of semantic attributes. Semantic maps have been utilized successfully in the areas of localization [41,42] and the prediction of pedestrian motion [43]. This approach is advantageous because it enables the representation of inherent distribution information within a discrete space while simultaneously filtering out noise. Our current work builds on this technique by applying it to the creation of semantic maps while additionally incorporating prior information from LiDAR’s intensity channel. As a result of this integration, we can generate semantic maps that are more stable given potentially noisy semantic images.

## 3. Materials and Methods

Our model consists of three main components: semantic segmentation, semantic association, and semantic mapping. Figure 1 illustrates the overall architecture. To begin, semantic segmentation networks are used to predict semantic labels on 2D images. These labels are then associated with densified 3D point clouds. Finally, a probabilistic mapping process is applied to convert the distribution of observations to a single label on a per-map pixel basis. In the following section, we will provide a detailed description of each component.

### 3.1. Image Semantic Segmentation

The first component, semantic segmentation, extracts the semantic labels from 2D images using neural networks. For each pixel in an image with shape *W* by *H*, the output is a label *c* from a set of predefined semantic classes *C* such as road, lane mark, and sidewalk. We offer two different segmentation network options: DeepLabV3Plus [6] and Hierarchical MultiScale Semantic Segmentation with HRNet+OCR (MScale-HRNet) [17]. At inference time, DeepLabV3Plus is faster but noisier while MScale-HRNet is slower and more memory intensive but provides higher-quality segmentation. We discuss the tradeoffs of both methods in the context of the final generated semantic map in later sections.

For DeepLabV3Plus, the feature extraction backbone is a lightweight ResNeXt50 [44] pre-trained on ImageNet [45]. Compared to other backbones like ResNet101 [46], ResNeXt50 achieves the same mean Intersection over Union (mIoU) value but with fewer parameters and faster inference times. To further improve inference time while preserving performance, we also employ depth-wise separable convolution in our spatial pyramid and decoder layers, inspired by [6,47].

Our DeepLabV3Plus semantic segmentation network is trained on the Mapillary Vistas dataset [18], which contains a large number of pixel-level semantic segmented images with 66 different labels in autonomous vehicle scenarios. The mapillary vistas dataset was at the start of our study the most comprehensive pixel level labeled dataset and is still considered a viable basis for training. We reduce the labels to 19 essential classes for our driving environment by removing non-essential labels (e.g., snow) and merging labels with similar semantic meanings (e.g., zebra line and crosswalk). This decision is based on the observation that some classes do not appear in our test environment. The details of label merging are described in Section 4.2.

MScale-HRNet uses a much larger HRNet+OCR backbone [48,49,50] that utilizes object context to achieve higher performance for irregular semantic regions. Additionally, by utilizing multi-scale segmentation with attention [51], the network pulls larger area semantic features from smaller scale images and more refined semantic features from larger scale images. The fusing of different scales is carried out in a hierarchical manner, enabling the scales used at inference time to be changed without retraining. As such, one can modify the runtime and memory requirement by lowering or increasing the scales used (this does result in changes in model performance). Overall, it achieves better segmentation than DeepLabV3Plus, at the cost of higher computation requirements.

From a quantitative standpoint on the cityscapes test set, DeepLabV3Plus is capable of attaining an mIoU of 82.10% [6], while MScale-HRNet achieves an mIoU of 85.10% [17].

### 3.2. Point Cloud Semantic Association

The second component, semantic association, reconstructs a 3D scene with semantic labels. Given the semantic images from semantic segmentation, this is achieved by assigning depth to the semantic image. However, the depth information is often not readily available. Depth estimation from multi-view geometry relies on salient features, which can be prone to errors on the road or under challenging lighting conditions. Alternatively, LiDAR sensors can capture depth information, but their sparse resolution, typically with only a few optical channels (e.g., 16), can make it difficult to infer the underlying geometry in real time. To overcome this challenge, our method leverages centimeter-level localization [52] to extract small, dense regions from a previously built dense point cloud map. These regions are then projected into the semantically segmented image to retrieve depth information. Building a dense point map can be automated and only requires driving through the area once, making it much less expensive than human labeling.

Assuming the vehicle is localized with respect to a point cloud map $P_{g}$ with coordinate $X_{v}$, local point cloud $P_{l}$ is extracted within a max distance in each dimension in the local coordinates of the vehicle. The transformation from the local point map to the localizer (Velodyne LiDAR) ${}^{l}T_{m}$ is given by precise centimeter-level localization. We also calibrate the camera with respect to the LiDAR using a non-iterative method solution for the PnP method [53], to estimate their relative transformation ${}^{c}T_{l}$. Therefore, the extrinsic transformation between the camera and the points map frame ${}^{c}T_{m}$ is known.
(1)$${}^{c}T_{m}=\left({}^{c}T_{l}\right)\left({}^{l}T_{m}\right).$$

Thus, semantic information for a point $X_{m}\in P_{l}$ can be retrieved from the label of its projected points in image coordinates $x_{i}$.
(2)$$x_{i}=K\pi \left({}^{c}T_{m}\right)X_{m}$$
where K is the camera intrinsic matrix and π=[I|0] is the canonical projection matrix.

Finally, we assign the semantic label of pixel $x_{i}$ in the semantic image to the point $X_{m}$ to form a semantic point cloud.

### 3.3. Semantic Mapping

A point cloud with semantic labels is a useful representation of a scene’s 3D geometry, but it can be affected by sensor measurement noise and small semantic label fluctuations. To address this, we use a local or global probabilistic map, where the former provides dense semantic cues around the ego-vehicle and the latter automates the process of building HD maps. Both local and global maps use semantic occupancy grids, with the main difference being the reference frame. Our comparisons are performed in the global frame.

A local probabilistic map is a BEV representation in the body frame (rear-axle) of the ego vehicle. We construct it for a given frame using the semantic point cloud and update it when there is a significant change in the ego-vehicle’s pose. On the other hand, a global probabilistic map operates directly in the global frame without the need for map transformations. A visual comparison of the two is shown in Figure 2.

The semantic occupancy grid has height *H*, width *W*, and channels *C*, with each channel corresponding to a semantic class of the scene. The channels for a cell in the BEV map model the semantic class probability distribution. When constructing the semantic point cloud, we project it onto the grid using the *x* and *y* components. The point will be associated with the nearest cell $c_{ij}$, which covers a d×d square area of the physical world. Then, we will update the channels in the cell based on the semantic label of the point.

We enhance the robustness of the semantic occupancy grid estimation using a probabilistic model that incorporates both the semantic and LiDAR intensity information from the point cloud to reduce the prediction error. We denote the semantic label distribution across all the channels as $S_{t}$, the observed semantic labels as $z_{t}$, and the observed LiDAR intensity as $I_{t}$. Thus, the task is to estimate $S_{t}$ from past observations, i.e., the probability distribution of $P\left(S_{t}|z_{1:t},I_{1:t}\right)$. We assume that observed semantic labels and LiDAR intensity are conditionally independent given $S_{t}$ and follow the Markov assumption to update the semantic probability.
(3)$$\begin{array}{c} P\left(S_{t}|z_{1:t},I_{1:t}\right)=\frac{1}{Z}P\left(z_{t}|S_{t}\right)P\left(I_{t}|S_{t}\right)P\left(S_{t-1}|z_{1:t-1},I_{1:t-1}\right) \end{array}$$

We introduce a normalization factor Z and assume that $P\left(S_{t}|z_{1:t-1},I_{1:t-1}\right)$ is equivalent to $P\left(S_{t-1}|z_{1:t-1},I_{1:t-1}\right)$. To enable a more precise probabilistic update, we use a 2D confusion matrix M to model $P\left(z_{t}|S_{t}\right)$, where each element in the matrix represents the probability of label *i* being predicted as label *j*. Additionally, we model $P\left(I_{t}|S_{t}\right)$ as a prior function of the intensity of each class in the scene.

The confusion matrix models the uncertainty of the model evaluated on a dataset, which describes the prior probability of a label $z_{t}$ being observed when the true class is $S_{t}$. As a result, for any point projected to the cell, all channels in the cell will be updated according to the confusion matrix. To ensure numerical stability, we use the logarithmic form to update the channels.

The intensity data collected by LiDAR sensors provides valuable information about different materials in the scene. For instance, the top image in Figure 3 shows a BEV intensity map of a road segment where lane markings appear brighter due to their high reflectivity. We use a threshold value *k* to segment out the lane markings and employ this information as a prior to better understand the layout of the scene. This approach can be especially helpful when semantic segmentation fails to capture the correct label due to poor lighting conditions.

## 4. Experiments

We perform experiments to verify the effectiveness of the proposed semantic mapping pipeline. We introduce our vehicle platform in Section 4.1. Then, we discuss the training, hyperparameter, and result comparison of semantic segmentation networks in Section 4.2. The semantic mapping results with an ablation study and analysis are presented in Section 4.3. Lastly, we compare different depth association approaches for semantic mapping in Section 4.4.

### 4.1. Platform

We collected our experimental data using one of our autonomous cars, as described in [52]. This car is equipped with a 16-channel LiDAR and six cameras, arranged with two cameras on the front, one on each side, and two on the back, as depicted in Figure 4. We recorded data from the front left camera, LiDAR, and vehicle position by driving through the UC San Diego campus. The camera data was streamed at approximately 13 Hz, while the LiDAR scans were performed at approximately 10 Hz. By driving through the campus, we were able to gather data for various urban driving scenarios, including challenging situations such as navigating steep hills, intersections, and construction sites.

### 4.2. Image Semantic Segmentation

We have two semantic segmentation networks: MScale-HRNet and DeepLabV3Plus. The MScale-HRNet pre-trained model (https://github.com/NVIDIA/semantic-segmentation, accessed on 30 May 2023) produces a high-quality semantic mask with clean edges most of the time. For its high-quality results, we use the straight-out-of-the-box pre-trained 65-class model directly. On the other hand, the DeepLabV3Plus produces much noisier results. Therefore, we reduce the total classes from 65 to 19 and retrain the model. In this subsection, we describe the configuration for MScale-HRNet and the training process for DeepLabV3Plus.

#### 4.2.1. MScale HRNet+OCR Configuration

MScale-HRNet allows for flexible scale selection during inference time. We chose three scales at 0.25, 0.5, and 1.0 for our experiments. Typical experiments are carried out with scales of 0.5, 1.0, and 2.0 but require more than 11 GB graphics processing unit (GPU) memory for input size 1920 × 1440. We observe that even with the downsized scales, the network was able to produce much cleaner and more accurate results than DeepLabV3Plus. Testing on our own vehicle data showed good generalization.

#### 4.2.2. DeepLabV3Plus Training Dataset

Our training dataset consists of 18,000 images, while our validation dataset has 2000 images, both of which are obtained from the Mapillary dataset [18]. To optimize our training process, we merged similar categories such as terrain and vegetation, and different types of riders and pedestrians, into a single human category, and various types of crosswalks into a unified crosswalk class. We also combined traffic-sign-back and traffic-sign-front into a single traffic-sign category and merged bridge images into the building category.

To further improve the training dataset, we applied several data augmentation techniques, including random horizontal flips with a probability of 0.5, random resizing with a scale ranging from 0.5 to 2, and random cropping. Additionally, we normalized the images to a distribution with a mean of 0.485, 0.456, and 0.406 and a standard deviation of 0.229, 0.224, and 0.225.

Our experiments indicate that the Mapillary dataset is similar to our driving scenarios, and the extensive data augmentation during the training process helps improve DeepLabV3Plus generalization. We did not observe a significant drop in performance when testing the DeepLabV3Plus model on the UC San Diego campus.

#### 4.2.3. DeepLabV3Plus Hyperparameters

To train our DeepLabV3Plus network, we employ synchronized batch normalization [5] with a batch size of 16. The training process lasts for 200 epochs, utilizing eight 2080Ti GPUs with an input image size of 640 × 640. The network’s output stride is eight.

To optimize the training process, we use the stochastic gradient descent (SGD) optimizer and apply a polynomial learning rate policy [6,54]. Specifically, we set the base learning rate to 0.005 and the power to 0.9, with the learning rate decaying over time according to the formula $base\_lr\times {\left(1-\frac{epoch}{max\;epoch}\right)}^{power}$. We set the momentum to 0.9 and the weight decay to 4 × 10${}^{-5}$.

#### 4.2.4. Comparison of Semantic Segmentation

We use the mIoU metric to assess a network’s performance. In the reduced 19-class Mapillary validation set, ResNeXt50 achieves an mIoU of 68.32%. Although its performance is slightly lower than that of ResNet101, ResNeXt50 requires significantly less memory (from 367 MB to 210 MB), making it more suitable for our onboard hardware with limited memory. We evaluate MScale-HRNet on the 65 class Mapillary validation set, where it achieves an mIoU of 59.71% for 65 classes.

Qualitatively, as shown in Figure 5, the two semantic segmentation networks perform similarly in close range with only one major difference. MScale-HRNet fills in gaps on the dash lane, while DeepLabV3Plus does not do this as consistently. This stems from irregular labeling in the Mapillary dataset, which we discuss further in Section 5.3. For our ground-truth labels, we do not fill in dash lanes, and that can lead to a performance drop for the MScale-HRNet approach.

The DeepLabV3Plus generates noisier results on the edges of the segments. MScale-HRNet outputs are cleaner with smooth edges. For areas further away from the camera, MScale-HRNet results give more details. However, these areas are not utilized since we clip the point cloud with a maximum distance to reduce error (see Section 4.3.4).

For an image size of 1920 by 1440, DeepLabV3Plus’ inference time is approximately 0.48 s per image and MScale-HRNet’s inference time is approximately 1.23 s per image when running on an NVIDIA GeForce RTX 2080Ti graphics card.

### 4.3. Semantic Mapping

We evaluate the quality of our map generation results by selecting a 1.1 km region of the UC San Diego campus, which has been manually annotated with an HD map containing road information, including crosswalks, sidewalks, and lane marks. The semantic map we generate has five channels—road, crosswalk, lane marks, vegetation, and sidewalk—with a resolution of d=0.2 m. Generating an accurate HD map requires considerable effort, but it demonstrates the value of automating the process.

#### 4.3.1. Metric for Semantic Mapping

In our initial work [19], we used mIoU and pixel accuracy as evaluation metrics. However, a direct comparison on IoU for lane marks is very sensitive to localization error. In Figure 6, we show the generated semantic map, ground truth, and disparity between the lane labels in these two maps. It can be seen that there are relatively consistent detections of the lane lines in the generated semantic map; however, when compared to ground truth they are off by 1 to 2 pixels (0.2–0.4 m since 1 px = 0.2 m). Given that the ground-truth lane is about 1 to 2 pixels wide, the offset leads to a very low true positive rate.

This offset is also egocentrically consistent across the entire generated map, leading us to believe this is a systematic problem unrelated to the semantic mapping approach. The offset can potentially be caused by an error introduced by the calibration between LiDAR and the camera, the asynchronous camera and LiDAR, the BEV conversion, or a discretization error in mapping.

For generating HD maps, the offset that is present is non-ideal. There are other tasks, however, which are less sensitive to this offset. An example is to use the map as a prior to provide context for scene understanding. The semantic map can be used in downstream tasks such as trajectory generation or motion prediction. In these scenarios, the existence of the semantic information is more important and centimeter-level mapping requirements may be too strict.

Therefore, in addition to IoU, we propose a metric to evaluate the performance of the semantic map that is tolerable to minor offsets. The proposed metric included a modified version of precision and recall. We dilate the ground truth to evaluate the precision of the generated map. We dilate the generated map for each label to evaluate the recall against the original ground truth. Specifically, we used a kernel size of 3, which tolerates a 20 cm error. We notice that these additional metrics match our observation of the performance of the model and thus can better guide our decision in hyperparameter tuning and model comparison.

Additionally, it is worth noting that the sparsity of the LiDAR point cloud may influence these metrics since the output may be accurate, but it may contain unclassified cells (holes). We mitigate this problem by using a smoothing kernel to interpolate the missing labels on our map.

#### 4.3.2. Modeling of Observation Uncertainty

To start, we verified the design of the confusion matrix M to model the uncertainty in the semantic segmentation stage. We explored two approaches for this purpose. The first approach, referred to as Vanilla, is defined by $\mu \left(I+\lambda 1\right)$, where λ is a hyper-parameter and μ is a normalization factor. The second approach is CFN, which is the confusion matrix of the semantic segmentation network in the Mapillary validation data set. During inference, we assigned each cell to the label with the highest probability. We present the quantitative results in Table 1. Our findings reveal that CFN significantly outperforms the Vanilla model in terms of IoU and recall, particularly for crosswalks and lane marks. The result is consistent across both backbone networks. This suggests that utilizing the confusion matrix of the network to model the prediction error in semantic segmentation leads to improved map generation results.

#### 4.3.3. Integration with LiDAR Intensity

To take advantage of the varying reflectivity of different road materials, we begin by filtering out all intensity data that falls below the normalized threshold value of k=14, which we manually calibrated for the Velodyne VLP-16 LiDAR (as shown in Figure 3). During the semantic mapping process, when our model predicts the presence of lane marks, we increase the logarithmic probability of that label by a constant factor γ. This suppresses our prediction of other classes and increases our confidence in predicting lane marks. In Table 1, the models that incorporate intensity data are denoted with a “+I” label. Comparing Vanilla+I to Vanilla, we observe improved accuracy and IoU scores for lane marks, but a slight decrease for roads and crosswalks, suggesting the benefit of integrating intensity data for lane mark prediction. However, this trend is not replicated for CFN+I compared to CFN, indicating that a more sophisticated function may be needed to model LiDAR intensity for further improvement.

#### 4.3.4. Effect of Clipping Range

We conduct experiments in Table 1 by clipping the local dense point maps extracted up to 10 m along the longitudinal axis and −15 to 15 m along the lateral axis of the vehicle because the semantic segmentation performance decreases significantly beyond this range. The effect of range on the final mapping result can be seen in experiments varying the clipping distance, summarized in Table 2.

The result suggests that a shorter distance yields better mapping performance for the most challenging lane mark class. We observed a similar pattern during the hyperparameter tuning for DeepLabV3Plus-based semantic mapping in our initial work [19] and believed that it was caused by a combination of reduced calibration error and more accurate semantic segmentation for closer ranges. We notice, however, that MScale-HRNet produces strong semantic segmentation for longer ranges but still exhibits the same trend. This leads us to believe that long-range mapping error is mainly related to camera calibration.

#### 4.3.5. Mapping Results

An example of the global map generated by our CFN+I DeepLabV3Plus model for the entire test region is shown in Figure 7. The figure highlights a region of the map, demonstrating our model’s ability to clearly capture and map the static elements of the road.

More examples from testing on the UC San Diego campus are shown in Figure 8. The first row shows results on more common environments such as intersections and road segments. The second row shows results on less structured environments such as parking lots and curved roads. In these cases, the pipeline can generate visually consistent semantic maps. The last row demonstrates noisy results in a construction zone, intersections with worn road markings, and uncommon road structures. Some of the issues can be addressed by leveraging vehicle-to-infrastructure communication [55].

### 4.4. Comparison with Different Depth Association Approach

Alternative methods to associate depth exist. In this section, we compare our approach, which leverages the dense point cloud map, with two approaches to associate depth, using sparse LiDAR scan and planar assumption.

#### 4.4.1. Comparison to Sparse LiDAR Scan

A potential alternative to associating semantic images with depth information is to utilize the real-time point cloud data generated by LiDAR. To accomplish this, we follow a similar mapping approach by projecting the point cloud onto the semantic image frame and constructing the semantic map. The real-time performance of this approach is demonstrated in Figure 9. However, due to the sparsity of point cloud scans from the 16-channel LiDAR used, constructing a semantic map at greater distances is challenging, particularly when the vehicle is moving at higher speeds. Therefore, to enable the creation of semantic maps for longer ranges with a sparse LiDAR, a pre-built dense point cloud map is necessary. With the advances in sensing technology, higher resolution or solid-state LiDARs such as a 128-channel LiDAR can potentially bridge the gap.

#### 4.4.2. Comparison to Planar Assumption

We also investigated a different approach, which involves back-projecting the 2D semantic image into 3D space using a homography, assuming a flat ground. This method eliminates black holes in the generated map. However, this approach is not suitable for urban driving scenarios with steep hills or road intersections, as illustrated in Figure 10, since the planar assumption fails under these conditions. Consequently, significant distortion occurs at longer ranges.

## 5. Discussion

We proposed a semantic mapping pipeline that leverages the semantic information from the image and geometric information from the point cloud to generate a probabilistic map in the BEV.

Our experiments highlighted the benefits of a probabilistic approach, which allowed us to capture fine details such as lane marks more accurately, in lieu of semantic segmentation noise. Additionally, we reviewed the appropriateness of mIoU as a mapping performance metric and argued that modified recall and precision better characterize pipeline performance (more details are provided in Section 5.1).

The semantic map generated by our pipeline can provide a rich context for downstream tasks. This includes direct use cases for navigation tasks and behavior prediction that requires semantic information to understand the underlying road geometry. For example, in recent work [56,57], a strategy for dynamic trajectory generation for urban driving was proposed. The methods leverage conditional generative models to align coarse global plans to local semantic maps and dynamically regress egocentric trajectories. The semantic features provided by our map can additionally be used as a base for HD Map generation. Combining the dense semantic map from our proposed pipeline with road network topology from approaches such as TopoNet [15] could provide both context and navigation cues, respectively. With these aforementioned potentials, the semantic mapping results can still be improved in many aspects.

### 5.1. IoU and Localization Error

Our analysis suggests that mapping IoU is highly sensitive to localization, and even a minor deviation causes the metric to underrepresent our results. As such, we present the results using metrics that are more tolerant to minor offsets in predictions, which are more consistent with our observations. While localization can be improved, since lane marks are typically 10 cm wide (https://safety.fhwa.dot.gov/roadway_dept/night_visib/pavement_marking/ch3.cfm, accessed on 30 May 2023) even perfect segmentation with 5 cm localization error reduces the IoU to 33%. As such, it is necessary to introduce additional metrics to represent results within an offset tolerance. Recent works [7,13,14,15] using vectorized representations are evaluated with different metrics that are not sensitive to localization error.

We notice the mapping offset error is consistent in the egocentric frame, which leads us to believe this is a systematic error of our equipment and not the semantic mapping pipeline. We believe that better calibration and sensor synchronization can improve mapping results by reducing the offset.

### 5.2. Semantic Segmentation

Another challenge is the robustness of semantic segmentation. The semantic segmentation model degrades in challenging lighting conditions and unseen environments. For example, as shown in Figure 11, the images may appear over-exposed and trees will cast a shadow on the road on a sunny data. In these scenarios, it is hard to correctly segment the lane marks. Additionally, road constructions and drivable regions that are not well painted compared to the normal road often confuse the network, leading to noisy segmentation.

We observe in our ablation study that considering LiDAR intensity values during predictions yields improvements in our performance. VectorMapNet [13] exhibits similar findings, where fusing LiDAR information boosts its performance in challenging environment conditions (puddles on the road). It is clear that multi-sensor approaches increase robustness. In the case of Figure 11, our multi-sensor approach fails to capture the lane mark in the final generated global map. Stronger semantic segmentation modules that consider temporal or spatial context are needed. They should be able to handle visual gaps in lane markings, whether due to wear or lighting conditions.

### 5.3. Mapillary Inconsistency

Another notable issue is the consistency of labels across the dataset used for both semantic segmentation networks. As we mentioned in Section 4.3.1, the Mapillary dataset [18] irregularly fills the dash lanes. For example, as shown in Figure 12 we can see dashed lanes being turned into solid lines in the first example, and in the second example a more complicated zebra-style lane region turned into a fully solid lane label. In the third example, however, the dashed line stays dashed.

We believe that this inconsistency in training data causes the networks to become confused and be more biased towards filling in gaps between lane marks that it finds appropriate. As seen in Figure 5, MScale-HRNet is more consistent in filling in the gaps than DeepLabV3Plus. We hypothesize that MScale-HRNet, being a more advanced network, has a greater ability to learn to fill in (as biased by Mapillary) than DeepLabV3Plus.

This has different ramifications on downstream task performance, as the resulting mapping is affected by the semantic segmentation filling behavior. For navigation tasks, maintaining dashed lanes is important for contextual understanding. Conversely, Zhou et al. [12] use particle filters for road network extraction, where filled lanes would be beneficial.

### 5.4. Disappearing Lanes and Discretization

The final major issue we observed is under-representation of semantic labels when mapping. Semantic image outputs (especially for MScale-HRNet) show consistent segmentation for lane lines; however, these lines do not necessarily transfer to the final map. By employing a confusion matrix, we account for semantic segmentation error, but we do not account for mapping discretization error.

The experiments ran in this paper were limited to a pixel resolution of 0.2 m due to memory constraints. This is a large area relative to a lane line’s standard width of 0.1 m. As such, a 0.2×0.2 region that should have been mapped to a lane line may have more road observations that lane line observations. In essence, as a result of our discretization size, this can cause lane line cells to be suppressed by surrounding road observations in the same cell.

An obvious fix would be to increase the discretization resolution; however, this comes with multiple problems. In addition to increased memory usage, it requires higher density in observations. Our current maps at 0.2 have holes due to the sparsity of a 16-channel LiDAR point cloud at driving speed. Thus, to counteract this either higher channel LiDARs, slower driving speed, or higher interpolation would be required. Potential exploration could be carried out by observing distributions of lane line points within a cell, to decide if it represents a lane line or noise. Additionally, discretization could be dropped completely by utilizing vector representations [7,13,14,15] for lanes instead, which are updated by lane observations.

## 6. Summary

By incorporating rich information from semantic labels on image frames, our method effectively introduces a statistical approach for identifying road features and mapping them in BEV, as demonstrated by our comparisons to manually annotated maps. This approach can be extended to automate HD map annotation for crosswalks, lane markings, drivable surfaces, and sidewalks, as well as incorporate center lane identifications for path tracking algorithms.

To address the scalability drawbacks of HD maps, future work will involve accounting for road network junctions and forks, allowing for the full automation of road network annotations leveraging graphical methods. While a combination of the proposed techniques may address the scalability and maintenance cost associated with dense point cloud maps for localization, it also opens up new areas of research in high-level dynamic planning. By dynamically estimating drivable surfaces, traffic lanes, lane markings, and other road features, centimeter-level localization may become unnecessary as long as immediate actions can be extracted from a high-level planner. In our future work, we plan to seek solutions for fully automating the HD mapping process while exploring the possibility of dynamic planning without a detailed dense point cloud map.

## Figures and Tables

**Figure 1 sensors-23-06504-f001:**
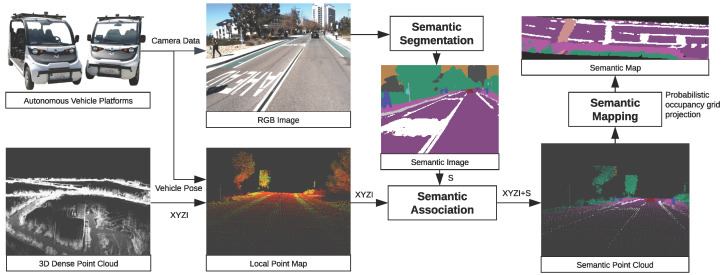
Our semantic mapping pipeline generates semantic labels for images, associates the labels with the local point cloud, and updates the semantic map in bird’s-eye view (BEV) probabilistically.

**Figure 2 sensors-23-06504-f002:**
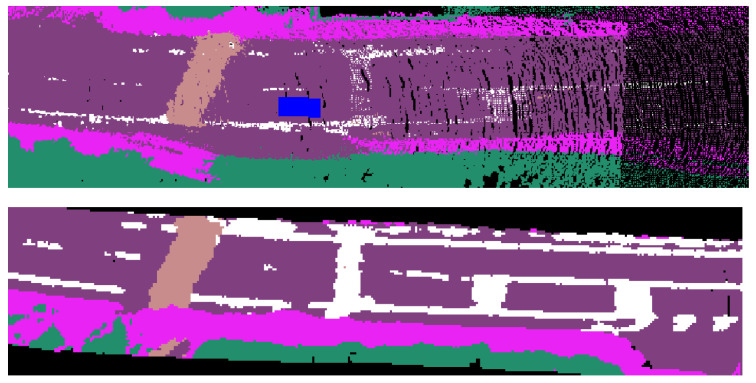
Top to bottom, local probabilistic map where blue car is ego vehicle; the same region in a final generated global map.

**Figure 3 sensors-23-06504-f003:**
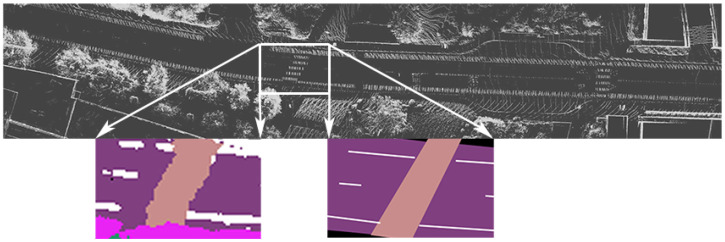
A visualization of our generated map (**bottom left**), the ground-truth label (**bottom right**), and the intensity thresholded LiDAR point cloud map (**top**).

**Figure 4 sensors-23-06504-f004:**
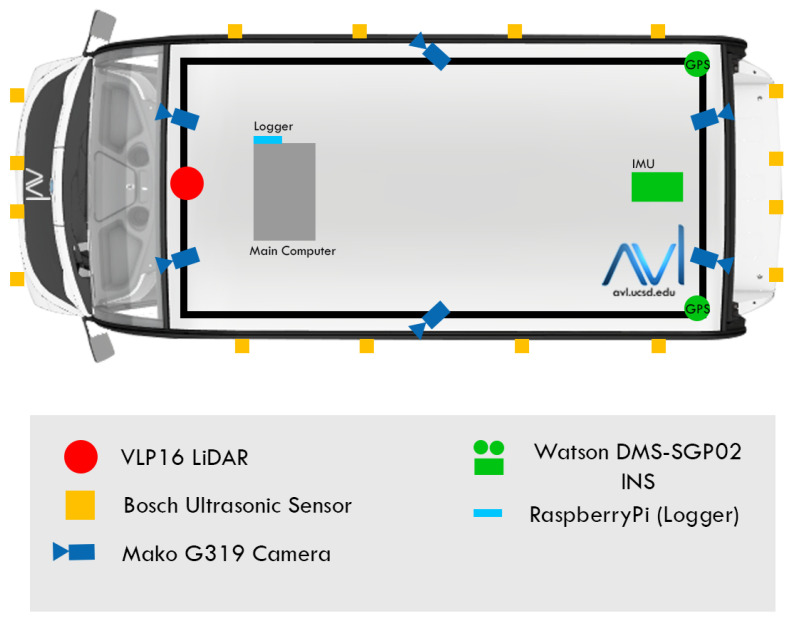
Vehicle sensor configuration.

**Figure 5 sensors-23-06504-f005:**
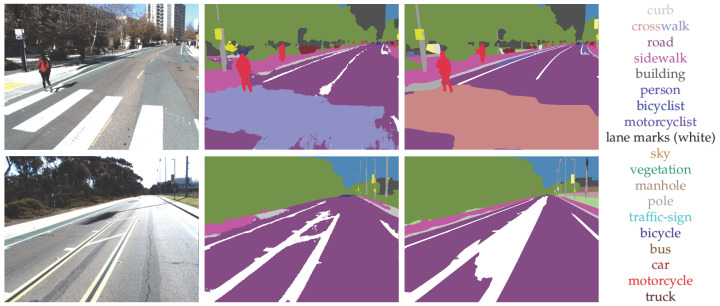
Semantic Segmentation Comparison (left to right, AVL Dataset Image, DeepLabV3Plus, and MSCale-HRNet) with labels colored correspondingly.

**Figure 6 sensors-23-06504-f006:**
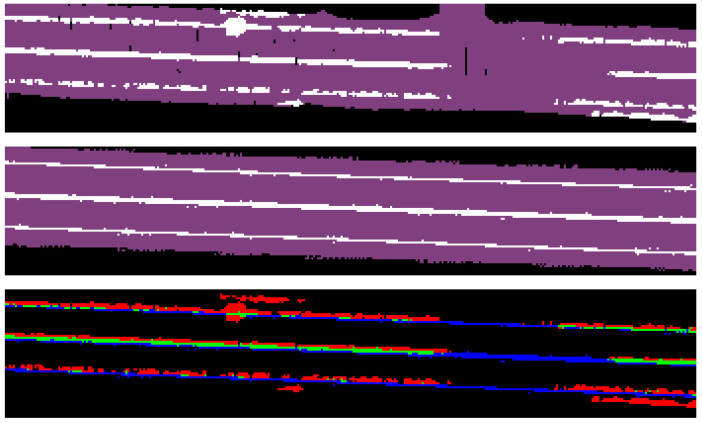
(From top to bottom) generated semantic map, ground truth, and disparity between lane labels. Green represents true positive, red represents false positive, and blue represents false negative.

**Figure 7 sensors-23-06504-f007:**
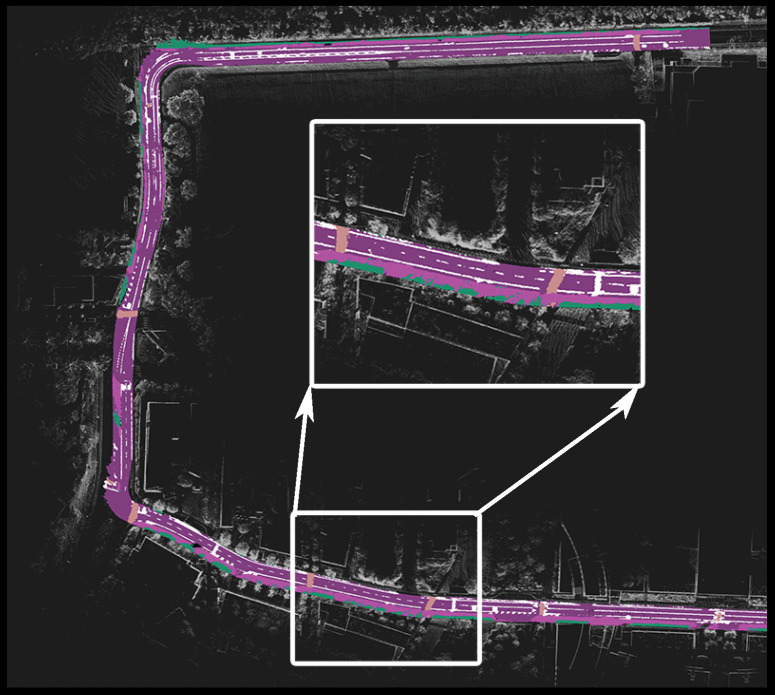
Generated map of testing data set in BEV, displayed on top of the dense point cloud map.

**Figure 8 sensors-23-06504-f008:**
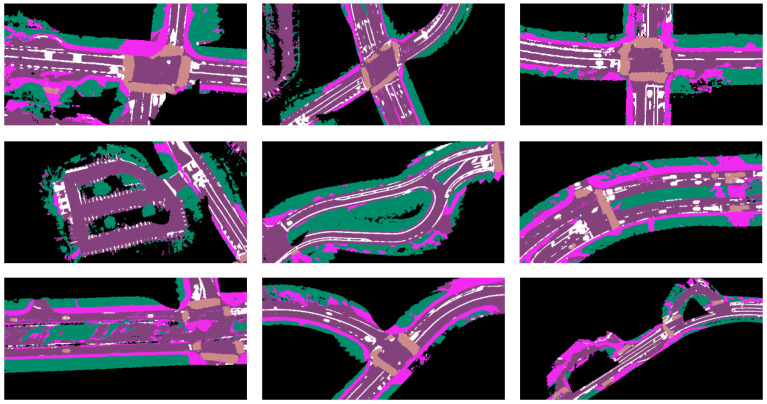
From top to bottom: results in structured environments, unstructured environments, and noisy results.

**Figure 9 sensors-23-06504-f009:**
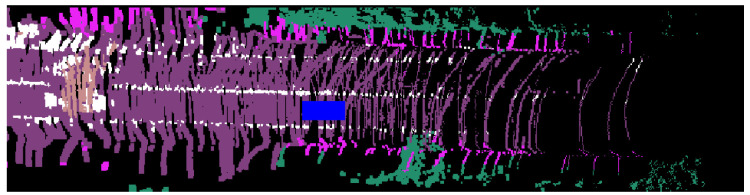
Semantic map generated from real-time LiDAR scan (black denotes areas not covered by LiDAR).

**Figure 10 sensors-23-06504-f010:**
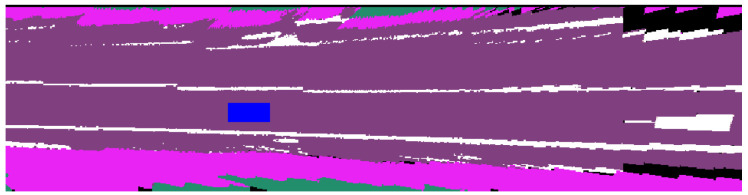
Semantic map generated by back-projecting 2D semantic image with a 3D planar assumption.

**Figure 11 sensors-23-06504-f011:**
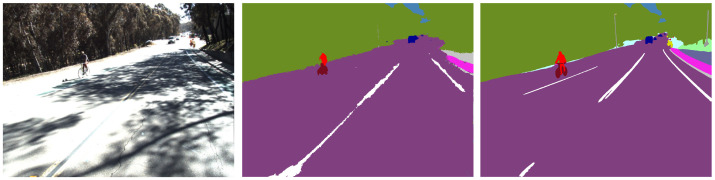
Semantic segmentation degradation from challenging lighting conditions. Left to right: original image, DeepLabV3Plus, and MScale-HRNet.

**Figure 12 sensors-23-06504-f012:**
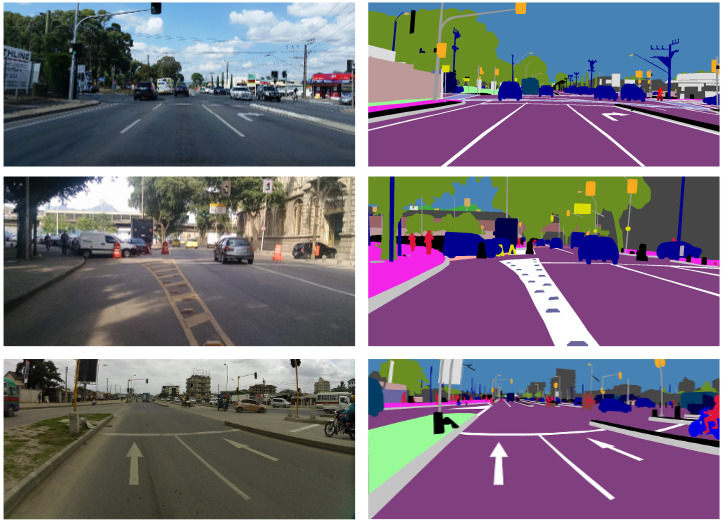
On the left we show images from the Mapillary dataset and on the right the visualized labels.

**Table 1 sensors-23-06504-t001:** Quantitative evaluation on our labeled data for road, crosswalk, and lane mark regions. Refer to Section 4.3 for details.

Network	Config	Road			Crosswalks			Lane Marks		
		Precision *	Recall *	IoU	Precision *	Recall *	IoU	Precision *	Recall *	IoU
4*DeepLabV3+	Vanilla	0.975	**0.786**	**0.678**	**0.990**	0.687	0.567	**0.762**	0.498	0.186
	Vanilla+I	0.975	0.784	0.674	**0.990**	0.677	0.552	0.757	0.576	0.213
	CFN	0.985	0.760	0.641	0.954	**0.745**	**0.622**	0.730	0.833	**0.335**
	CFN+I	0.985	0.759	0.640	0.954	0.741	0.616	0.727	**0.835**	**0.335**
4*MScale-HRNet	Vanilla	0.983	0.771	0.674	0.911	0.658	0.519	0.725	0.451	0.191
	Vanilla+I	0.984	0.770	0.670	0.909	0.646	0.502	0.720	0.522	0.207
	CFN	**0.989**	0.758	0.647	0.897	0.697	0.547	0.752	0.807	0.320
	CFN+I	**0.989**	0.757	0.645	0.892	0.690	0.537	0.749	0.810	0.321

* The precision and recall are not in common definition. See Section 4.3.1 for details. Bolded font indicates the best result.

**Table 2 sensors-23-06504-t002:** Ablation study on point map maximum clipping distance.

Range	Road			Crosswalks			Lane Marks		
	Precision *	Recall *	IoU	Precision *	Recall *	IoU	Precision *	Recall *	IoU
30	0.985	**0.847**	0.702	0.695	0.760	0.495	0.555	0.567	0.182
15	**0.989**	0.836	**0.706**	0.823	**0.766**	**0.560**	0.683	0.761	0.270
10	**0.989**	0.757	0.645	**0.892**	0.690	0.537	**0.750**	**0.810**	**0.321**

* The precision and recall are not in common definition. See Section 4.3.1 for details. Bolded font indicates the best result.

## Data Availability

The data and code presented in this study will be made openly available in https://github.com/AutonomousVehicleLaboratory/semantic_mapping_v2 (accessed on 30 May 2023).

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
