# Peer review of "Probabilistic Semantic Mapping for Autonomous Driving in Urban Environments"

_sensors, 2023, doi:10.3390/s23146504_

Round 1
Reviewer 1 Report
1. The lane markings within the intersection are very blurry. Why? can it be improved?
2. This research actually deal with semantic mapping rather than semantic SLAM, since as object positions are not considered.
3. On page 3, section 2.2, the authors claim that exisiting studies have certain limitations regarding the use of specific types of LiDAR technology However, the provided citation does not accurately correspond to the source material.
4.There are inconsistencies between the citations mentioned in the main text and those listed in the reference section. I recommend thoroughly reviewing both sections to ensure consistency and accuracy.
Minor editing of English language required
Author Response
1) The lane markings within the intersection are very blurry. Why? can it be improved?
Response: In figure 8 (revised version), we show in the last row that some of the intersection crosswalk markings are blurry. They are either due to construction blocking the view or worn road marking that lacks maintenance. We made it clear in the revised version that it is due to worn road markings. In the discussion, we added that further improvements require “Stronger semantic segmentation modules that consider temporal or spatial context are needed. They should be able to handle visual gaps in lane markings, whether due to wear or lighting conditions.”
2) This research actually deal with semantic mapping rather than semantic SLAM, since as object positions are not considered.
Response: Our initial version used the word localization a few times which should actually imply mapping. We change the wording from localization to mapping (or similar) when appropriate to better describe methods/results.
3) On page 3, section 2.2, the authors claim that exisiting studies have certain limitations regarding the use of specific types of LiDAR technology However, the provided citation does not accurately correspond to the source material.
Response: Initially, we used the first name instead of the last name to refer to the authors. We modified the citation to match the correct paper.
4) There are inconsistencies between the citations mentioned in the main text and those listed in the reference section. I recommend thoroughly reviewing both sections to ensure consistency and accuracy.
Response: We double-check the citation to make sure they are consistent as referenced.
Reviewer 2 Report
The authors of this article propose a mechanism to improve the automatic generation of dense probabilistic semantic maps in urban driving environments. Thus, to do so the authors employ dense maps from a 16-channel LiDAR (Light Detection and Ranging) and semantically labeled images produced by deep neural networks. Therefore, the approach is able to identify roads, crosswalks, and sidewalks. The proposal is tested in an urban area where segmentation networks generate a semantic map with road features.
The related work is concisely presented by describing semantic segmentation, semantic mapping, HD map generation, and probabilistic map. The latter is the approach followed by the authors. The aim is to obtain more stable semantic maps facing noisy semantic images. This point of the paper should be improved since the authors only say in Section 2.4 that such a technique has been successfully used in applications of localization, and prediction of pedestrian motion. I believe that more should be said about this type of technique since it is the one followed by the author's proposal.
Next, the authors describe the three components of their model: image semantic segmentation, point cloud semantic association, and semantic mapping. This last subsection presents an expression to update the semantic probability, which is the only equation in the whole paper.
The authors employ two semantic segmentation networks: MScale-HRNet and DeepLabV3Plus. Then, a quantitative evaluation is carried out on the labeled data for road, crosswalk, and lane mark regions. Precision, recall, and IoU metrics have been considered. Thus, it is shown that high precision can be attained with the proposed approach. At the end of the article, the authors discuss some environmental variables that may impact the results and provide an argument for that.
The paper is well written; there are a few minimal English details that must be corrected. An aspect that must be further improved is the citations to figures, which do not follow an order. The figures in the article are cited as follows: Figure 1, 6, 10, 2, 3, 4, 10, 5, 7, 8, 9, 11, 12, 3. This may distract the reader and must be improved by providing a better-structured article for fluent reading.
One very nice thing is that the authors make their code available on GitHub. Thus, a researcher working on the subject would be potentially able to reproduce the results provided in the article.
Finally, this is an extension of an IEEE conference paper published in 2020, where the authors considered mIoU and pixel accuracy as metrics. The additional metrics considered in this work show that CFN provides better results than the Vanilla model in IoU and recall. The last paragraph of the Introduction section says that this is an extension of the conference paper by adding more experiments and detailed analysis. Thus, I suggest improving such a sentence by clearly specifying what is the impact of extending this work in comparison with the conference paper. The reader has to dig out into the paper to have a more clear insight into such improvements. Section 3.3 (Semantic Mapping) might be clearly improved by providing a more detailed analysis.
Section 2.3, line 100 says "[7] propose to generate ...", should be written "[7] propose to generate ...".
Line 189, page 5 says "The point will be associate ...", should be written as "The point will be associated ..."
Line 465, page 15 says "Our current maps at 0.2 has ...", please correct.
Author Response
The authors of this article propose a mechanism to improve the automatic generation of dense probabilistic semantic maps in urban driving environments. Thus, to do so the authors employ dense maps from a 16-channel LiDAR (Light Detection and Ranging) and semantically labeled images produced by deep neural networks. Therefore, the approach is able to identify roads, crosswalks, and sidewalks. The proposal is tested in an urban area where segmentation networks generate a semantic map with road features.
The related work is concisely presented by describing semantic segmentation, semantic mapping, HD map generation, and probabilistic map. The latter is the approach followed by the authors. The aim is to obtain more stable semantic maps facing noisy semantic images. This point of the paper should be improved since the authors only say in Section 2.4 that such a technique has been successfully used in applications of localization, and prediction of pedestrian motion. I believe that more should be said about this type of technique since it is the one followed by the author's proposal.
Response: We’ve added related literature to provide a background of probabilistic maps.
Next, the authors describe the three components of their model: image semantic segmentation, point cloud semantic association, and semantic mapping. This last subsection presents an expression to update the semantic probability, which is the only equation in the whole paper.
Response: We additionally formalize the semantic segmentation and semantic association part by incorporating the projective geometry equations in the semantic association step.
The authors employ two semantic segmentation networks: MScale-HRNet and DeepLabV3Plus. Then, a quantitative evaluation is carried out on the labeled data for road, crosswalk, and lane mark regions. Precision, recall, and IoU metrics have been considered. Thus, it is shown that high precision can be attained with the proposed approach. At the end of the article, the authors discuss some environmental variables that may impact the results and provide an argument for that.
The paper is well written; there are a few minimal English details that must be corrected. An aspect that must be further improved is the citations to figures, which do not follow an order. The figures in the article are cited as follows: Figure 1, 6, 10, 2, 3, 4, 10, 5, 7, 8, 9, 11, 12, 3. This may distract the reader and must be improved by providing a better-structured article for fluent reading.
Response: We moved figures closer to their first reference in main text, and as such the references are better placed and ordered. The figure reference order is now: 1, 2, 3, 4, 5, 6, 3, 7, 8, 9, 10, 11, 12, 5.
One very nice thing is that the authors make their code available on GitHub. Thus, a researcher working on the subject would be potentially able to reproduce the results provided in the article.
Finally, this is an extension of an IEEE conference paper published in 2020, where the authors considered mIoU and pixel accuracy as metrics. The additional metrics considered in this work show that CFN provides better results than the Vanilla model in IoU and recall. The last paragraph of the Introduction section says that this is an extension of the conference paper by adding more experiments and detailed analysis. Thus, I suggest improving such a sentence by clearly specifying what is the impact of extending this work in comparison with the conference paper. The reader has to dig out into the paper to have a more clear insight into such improvements. Section 3.3 (Semantic Mapping) might be clearly improved by providing a more detailed analysis.
Response: The contributions are clarified at the end of the introduction. The contributions include new semantic segmentation models, open-sourcing the pipeline, and analysis for new metrics (precision and recall)
We've also corrected the English language issues.
Reviewer 3 Report
The paper establishes a statistical approach for identifying and localizing road features in the process of semantic mapping. The subject of the paper is very interesting, contemporary, and in line with the aims and scope of the Journal. The paper is well-structured and well-written. It provides an original approach and useful results. However, some minor issues need to be resolved.
1. The authors mentioned the research gap in the introduction, but they should explicitly state the identified research gaps based on the related work that they reviewed.
2. This is a very contemporary research field that is developing very fast. However, the authors referred to only several papers from the past few years. The authors should consider updating their related work section with more recent papers. There are plenty of papers dealing with this topic only in 2023.
3. The authors should present the methodology in more detail. They provided a general architecture in Figure 1, but this architecture should be presented in a more engineering and scientific way (provide algorithms for the software you developed).
4. The authors should highlight what is the novelty of their paper and software, and what was already established earlier.
5. The authors should make a better connection of the discussion with the related work that they reviewed.
6. The discussion should highlight the main theoretical and practical implications of the paper.
7. There are certain technical issues:
a) There should be at least a couple of sentences between headings of different levels (e.g. between section 2 and sub-section 2.1, etc.).
b) Please keep the figure captions as short and informative as possible. The captions are too extensive. Write everything you need about the figures in the main text and then put e reference to a figure.
c) Figures should be placed as close as possible to the place where they are first mentioned in the main text. For example, you give Figure 1 in the introduction and refer to it as far as in section 3.
d) Table captions should be written above the table, not below. Check the Table 1. Also, this table is not formatted according to the instructions for authors.
e) References in the reference list are not formatted according to the instructions for authors (e.g. journal names should be abbreviated).
f) Acronyms/Abbreviations/Initialisms should be defined the first time they appear in each of three sections: the abstract; the main text; the first figure or table. For example, “HD” and „ROS“ are not defined in the abstract. Check the rest of the paper.
g) Equations should be numbered.
Author Response
The paper establishes a statistical approach for identifying and localizing road features in the process of semantic mapping. The subject of the paper is very interesting, contemporary, and in line with the aims and scope of the Journal. The paper is well-structured and well-written. It provides an original approach and useful results. However, some minor issues need to be resolved.
1. The authors mentioned the research gap in the introduction, but they should explicitly state the identified research gaps based on the related work that they reviewed.
Response: We restructure the introduction to connect the related work and gap. To summarize, related research is either only focused on semantic understanding which doesn’t account for mapping, or use the aerial image / high-cost sensors that limit data availability, or generates lane-level HD map with limited features. Our method proposes a full mapping pipeline that goes beyond single-frame semantic understanding, We use a point cloud map that can be built from a sparse 16-channel LiDAR. And we leverage state-of-the-art semantic segmentation networks which are flexible and generates a rich semantic map. Additionally, we incorporate a confusion matrix and LiDAR intensity to map the lane marks.
2. This is a very contemporary research field that is developing very fast. However, the authors referred to only several papers from the past few years. The authors should consider updating their related work section with more recent papers. There are plenty of papers dealing with this topic only in 2023.
Response: We’ve updated the citations by adding most recent work up to CVPR 2023. For example, HDMapNet, VectorMapNet, MapTR and TopoNet.
3. The authors should present the methodology in more detail. They provided a general architecture in Figure 1, but this architecture should be presented in a more engineering and scientific way (provide algorithms for the software you developed).
Response: Additionally details have been added to the method section, especially for semantic segmentation and semantic association. We have additionally added more formal definitions and equations.
4. The authors should highlight what is the novelty of their paper and software, and what was already established earlier.
Response: We’ve adjusted the structure and wording in the introduction to highlight the novelty of our paper and software. We also explicitly reference the work that we are based on.
5. The authors should make a better connection of the discussion with the related work that they reviewed.
Response: Added snippets to the discussion to better connect our findings to related work. Mentioned how our pipeline for dense semantic maps can be combined with topological information from other cited approaches that can be a base for HD Maps. Noted that other cited approaches that utilize vectorized representations have a different performance metric that is not susceptible to localization error, unlike mIoU for mapping performance. Connected similar findings between our work and cited works on the benefits of incorporating LiDAR for semantic extraction. Commented on the pros and cons of filling dashed lines, and their potential effects on particle filtering for road network tasks, which is done in another cited work.
6. The discussion should highlight the main theoretical and practical implications of the paper.
Response: We’ve added additional discussion and rephrased the first subsection to highlight theoretical and practical implications.
There are certain technical issues:
a) There should be at least a couple of sentences between headings of different levels (e.g. between section 2 and sub-section 2.1, etc.).
Response: Added a couple sentences summarizing section 2, to add sentences between headings of different levels.
b) Please keep the figure captions as short and informative as possible. The captions are too extensive. Write everything you need about the figures in the main text and then put e reference to a figure.
Response: We’ve rewritten the figure captions to make them more concise.
c) Figures should be placed as close as possible to the place where they are first mentioned in the main text. For example, you give Figure 1 in the introduction and refer to it as far as in section 3.
Response: Figure placement and order have been rearranged so that figures are referred to as close as possible to where they are first mentioned.
d) Table captions should be written above the table, not below. Check the Table 1. Also, this table is not formatted according to the instructions for authors.
Response: Table 1 is in fact the caption for semantic segmentation and thus being merged into Figure 5.
e) References in the reference list are not formatted according to the instructions for authors (e.g. journal names should be abbreviated).
Response: We’ve added the field to BibTeX as required by the instruction for authors (Using ISO4 journal names, add the address and date of the conference and add doi when possible). The address and date of the conference are not shown using the overleaf template and we emailed the editor's office for guidance.
f) Acronyms/Abbreviations/Initialisms should be defined the first time they appear in each of three sections: the abstract; the main text; the first figure or table. For example, “HD” and „ROS“ are not defined in the abstract. Check the rest of the paper.
Response: Reviewed the entire paper for any acronyms, and ensured that they were defined prior.
g) Equations should be numbered.
Response: Added numbers for equations in the main text.
Round 2
Reviewer 1 Report
The authors have addressed all of my comments.
Author Response
Thank you for taking the time to give us constructive feedback!
Reviewer 2 Report
Thanks for updating your paper according to the suggestions in the previous round.
English has improved a lot in this version.
Author Response
Thank you for taking the time to give us constructive feedback.